# Human motor learning dynamics in high-dimensional tasks

**Ankur Kamboj** **[1]** *, **Rajiv Ranganathan[2], Xiaobo Tan[1], Vaibhav Srivastava[1]**

**1** Department of Electrical and Computer Engineering, Michigan State University, East Lansing, Michigan, United States of America, **2** Department of Kinesiology, Michigan State University, East Lansing, Michigan, United States of America

* ankurank@msu.edu

**Data Availability Statement:** The data and the code files necessary to reproduce the results in the paper can be found at the GitHub repository: https://github.com/nkur/HMLmodel.

## Abstract

Conventional approaches to enhance movement coordination, such as providing instructions and visual feedback, are often inadequate in complex motor tasks with multiple degrees of freedom (DoFs). To effectively address coordination deficits in such complex motor systems, it becomes imperative to develop interventions grounded in a model of human motor learning; however, modeling such learning processes is challenging due to the large DoFs. In this paper, we present a computational motor learning model that leverages the concept of motor synergies to extract low-dimensional learning representations in the high-dimensional motor space and the internal model theory of motor control to capture both fast and slow motor learning processes. We establish the model's convergence properties and validate it using data from a target capture game played by human participants. We study the influence of model parameters on several motor learning trade-offs such as speed-accuracy, exploration-exploitation, satisficing, and flexibility-performance, and show that the human motor learning system tunes these parameters to optimize learning and various output performance metrics.

## Author summary

Examining the learning and acquisition of motor skills in humans when facing complex, high-dimensional tasks is vital for understanding human motor learning, optimizing the performance of human-in-the-loop systems, improving learning outcomes, and facilitating rehabilitation. Toward this goal, we develop a normative model of human motor learning in high-dimensional novel motor tasks and show that it explains experimental data reasonably well. Further, through a model-based investigation, we examine various motor learning trade-offs, such as exploration-exploitation, speed-accuracy, satisficing, and flexibility-performance. These findings provide a foundational insight into how the human brain may balance these trade-offs during learning.

**Funding:** This work was supported by the NSF Grant CMMI 1940950 (https://www.nsf.gov/awardsearch/showAward?AWD_ID=1940950) received by the authors V.S., X.T., and R.R. The funders had no role in study design, data collection and analysis, decision to publish, or preparation of the manuscript.

**Competing interests:** The authors have declared that no competing interests exist.

## Introduction

Understanding human motor learning and relearning is of critical importance for domains such as motor skill acquisition and rehabilitation. Despite several advances in the theoretical mechanisms underlying motor learning [1], these mechanisms have been predominantly based on task paradigms that often do not consider the high dimensionality of the motor system at multiple levels, termed the 'degrees of freedom problem' [2]. Consequently, expanding existing models to investigate motor learning in high-dimensional motor tasks presents non-trivial challenges such as simultaneous control of large DoFs, and computational complexities arising from learning in high-dimensional spaces.

A key feature of learning in such high dimensional motor tasks is to handle several trade-offs such as the classic speed-accuracy trade-off [3], the exploration-exploitation trade-off [4], and the computational simplicity vs. flexibility trade-off [2, 5]. Moreover, cognitive models of decision-making have investigated phenomena such as satisficing [6] under such constraints, but these trade-offs have not been examined in motor learning contexts. In this work, we investigate these trade-offs by developing a model of motor learning in high-dimensional motor tasks. We leverage the idea that the human nervous system uses a small number of *motor synergies*, defined as a coordinated motion of a group of joints, to control and manipulate high DoF motor systems [7–9]. This allows us to extract low-dimensional learning representations in high-dimensional motor space, which enables the formulation of a computational model that can tackle the computational complexity of a large number of DoFs.

The approach of Ref [10] is closest to ours with the key distinction that [10] considers discrete tasks, while we focus on a continuous learning paradigm, which includes adaptation in the presence of continuous visual feedback. This shift in the experimental paradigm necessitates the reformulation of the model from first principles and the inclusion of a human motion perception model. Initial efforts towards motor learning on novel motor tasks in the presence of continuous visual feedback can also be found in [11] where heuristic methods are used for data fitting. Compared to the model proposed in [11], in the current work, we have a more refined model, a systematic data fitting procedure, a thorough investigation of various motor learning phenomena, and a bigger pool of experimental data to validate the efficacy of our model in explaining the motor learning behavior.

The goal of this work is to study human motor learning for high-dimensional novel learning task through a computational model that can explain various motor learning phenomena. Toward this end, we first develop an integrated dynamic model of human motor learning through the formation of internal representation, including models for perception, and forward and inverse learning. We use the motor synergies extracted from the human hand postures to create low-dimensional learning states, thus tackling the issue of increasing computational complexity with increasing DoFs of motor systems. We establish convergence properties of the proposed model and after fitting human participant data, we show that the proposed model can explain human motor learning and output performance behavior well. We then use the proposed model to systematically investigate the influence of model parameters on several motor learning trade-offs, including, speed-accuracy, exploration-exploitation, satisficing, and flexibility-performance. This analysis reveals how the motor system optimizes the use of synergies to control large degrees of freedom, how they manage various learning trade-offs, and how satisficing behavior is observed in a motor learning setting.

## Motor learning experiment

In our experiment, healthy participants learn a novel motor task by playing a target capture game [12]. Each participant wears a data glove, which records the movements of the 19 finger

joints. A body-machine interface (BoMI) then projects the 19-dimensional finger movements onto the movement of a cursor on a 2-D computer screen using a matrix. Specifically, the BoMI projects finger joint velocities $\boldsymbol{u} \in \mathbb{R}^m$ to cursor velocity $\dot{\boldsymbol{x}} \in \mathbb{R}^n$ using a matrix $C \in \mathbb{R}^{n \times m}$ such that

$$\dot{\boldsymbol{x}} = C\boldsymbol{u}. \tag{1}$$

Here, $n = 2$ (the 2-D computer screen) and $m = 19$ (the 19 finger joints). Let $\boldsymbol{q} \in \mathbb{R}^m$ be the vector of finger joint angles and thus, $\dot{\boldsymbol{q}} = \boldsymbol{u}$. Since the mapping is linear and the mapping matrix $C$ is time-invariant, the joint velocity to cursor velocity mapping, and the joint position to cursor position mapping are equivalent, as long as the initial cursor position is the same. Also note that this setup is different from the experimental paradigm undertaken in other studies, for instance [13], where the joint (IMU) angles are mapped to the end-effector velocities of a robot arm being controlled.

Participants first engage in a calibration phase where they move their hand fingers randomly while avoiding any extreme range of motion. The corresponding hand finger posture data is collected through the data glove, centered, and then PCA is performed to extract the principal components (PCs). The first two PCs are used as the rows to design the projection matrix $C$ specific to each participant. The game bounds are also specialized to each participant to make sure that all the points in the game window are reachable (refer to Methods for more details). During the target capture gameplay, the participants need to move the cursor to the prescribed target point one after the other displayed as the centers of the squares on a $5 \times 5$ square grid, and a new target is prescribed only after the current target is captured. The participants train on 4 targets (3 outer targets and 1 center target) for 8 sessions, each comprising of 60 target capture movements. The sequence of targets prescribed in each session is randomized but it always consists of 12 center out movements to each of the 3 outer targets from the center target, and 24 movements to the center target (8 from each of the 3 outer targets). Participants are further instructed to try and capture the prescribed targets within 2s of the movement onset from the previously captured target position, failing which the target square highlights in red to indicate that they have exceeded the time. Although there is no maximum time limit to capture the target, a scoring system based on the movement time and accuracy is shown on top of the game window for motivation purposes (refer to [12] for more details).

Through the gameplay, participants learn how to move various finger joints to make the cursor move along a desired trajectory, and in doing so they learn coordinated finger joint movements that are consistent with the projection matrix $C$ ($C$ is unknown to the participants). This experiment involves learning in high-dimensional human motor systems (the finger joint space), whereas the output performance feedback is in the low-dimensional screen space. Since the mapping $C$ from the 19-dimensional joint space to the 2-dimensional screen space is many-to-one, there are multiple solutions to the inverse kinematic mapping due to the large null space of $C$. This task is redundant in the sense that a desired cursor movement can be achieved with multiple synergistic motions of finger joints.

## Model

One way that motor learning in novel environments can occur is through the formation of internal representations, including models for perception, forward learning, and inverse learning [14–16]. Our formulation of these forward and inverse models follows the convention described in [10], which was originally introduced in [17].

There is also evidence that humans can control a high number of DoFs using a small number of coordinated joint movement patterns called synergies [7, 18–21]. We, therefore,

decompose the mapping matrix $C = W\Phi$, where $\Phi \in \mathbb{R}^{h \times m}$ is a matrix of $h$ basic synergies underlying coordinated human finger motions, and $W \in \mathbb{R}^{n \times h}$ represents the contributions (weights) of these synergies (see Methods: *Extracting motor synergies* for details on how the synergy matrix $\Phi$ is formed). Without loss of generality, the rows of $\Phi$ are assumed to be orthogonal, i.e., the synergies contributing to the hand motions lie in orthogonal spaces. While our motor learning model is in a high-dimensional space, these synergies reduce the size of the learning space and enable efficient learning by reducing the amount of exploration.

## Human perception model of BoMI

Since our experimental paradigm consists of humans learning under continuous feedback, we propose a perception model that explains how humans process continuous feedback signals. The BoMI mapping (Eq (1)) is described in terms of joint and cursor velocities. However, we hypothesize that humans interacting with the BoMI perceive these velocities as increments in cursor positions and finger joint angles. Consequently, we re-write (Eq (1)) using filtered joint and cursor position data [22] as

$$\delta \boldsymbol{x} = C \delta \boldsymbol{q}, \tag{2}$$

where $\delta \boldsymbol{x}$ and $\delta \boldsymbol{q}$ are termed *filtered increments in cursor positions*, and *filtered increments in joint angles*, respectively. Here $\delta \boldsymbol{q}$ evolves as per the dynamic equation

$$\dot{\delta \boldsymbol{q}} = -a\delta \boldsymbol{q} + \boldsymbol{u} + \boldsymbol{\xi}_q, \tag{3}$$

where the parameter $a$ controls the smoothing weights assigned to previous velocities, and $\boldsymbol{\xi}_q$ is the *perceptual noise* that captures the inaccuracies in human motion perception (refer to S1 File:Section 1.1 for a detailed derivation). We call $a$ the *perceptual recency parameter*.

## The forward learning model

Learning the forward model corresponds to learning the forward BoMI mapping matrix $C$, which maps the finger motions to the cursor motion. We represent the participant's implicit estimate of $C$ at time $t$ by $\hat{C}(t)$. Correspondingly, for the participant's change in finger joint angles $\delta \boldsymbol{q} \in \mathbb{R}^m$, the participant's estimated change in cursor position $\widehat{\delta \boldsymbol{x}} \in \mathbb{R}^n$ is determined by the forward mapping estimate $\hat{C}$:

$$\widehat{\delta \boldsymbol{x}} = \hat{C}\delta \boldsymbol{q} = \hat{W}\Phi\delta \boldsymbol{q}, \tag{4}$$

where $\hat{W}(t) \in \mathbb{R}^{n \times h}$ is the estimate of the matrix $W$. It represents the estimated weights that the human participant assigns to each synergy at time $t$. The learning space is therefore reduced from $\mathbb{R}^{n \times m}$ to $\mathbb{R}^{n \times h}$, a significant reduction, thus making our proposed model much more tractable. It follows from (Eqs (2) and (4)) that the estimation error

$$\epsilon = \delta \boldsymbol{x} - \widehat{\delta \boldsymbol{x}} = -\tilde{W}\Phi\delta \boldsymbol{q}, \tag{5}$$

where $\tilde{W}(t) = \hat{W}(t) - W \in \mathbb{R}^{n \times h}$ is called the parameter estimation error.

Applying the gradient descent on $\frac{1}{2}\|\epsilon\|^2$ with respect to $\hat{W}$ leads to

$$\dot{\hat{W}} = -\gamma \nabla_{\hat{W}} \frac{1}{2}\|\epsilon\|^2 = \gamma \epsilon \delta \boldsymbol{q}^\top \Phi^\top, \tag{6}$$

where $(\cdot)^\top$ represents the transpose and $\gamma > 0$ is the rate at which human participant learns the forward mapping. Accordingly, we call $\gamma$ the *forward learning rate*. We posit (Eq (6)), with $\gamma$ as

a tunable parameter, as a model for human forward learning dynamics. Similar models, whose dynamics evolve as a result of reducing some error metric, have been previously used in motor learning literature [10, 23], thus making our proposed model consistent with the error-based human motor learning paradigm.

### The inverse learning model

Learning the inverse model requires identifying the finger joint motions that are needed to drive the cursor on the screen to the right target by achieving the target hand posture.

Given the current cursor position $x$ and the desired cursor position $x^{\text{des}}$, we define $e_x = x^{\text{des}} - x$. We hypothesize that humans choose their nominal joint velocities to minimize the cost function

$$J(e_x, u) = \frac{1}{2}\left\|\hat{C}u - k_P e_x\right\|^2 + \frac{\mu}{2}\|u\|^2, \tag{7}$$

where the minima associated with the first term determines a $u$ that makes the human's estimate of cursor velocity $\hat{C}u$ equal to the error-driven proportional feedback $k_P e_x$, for some $k_P > 0$, and the second term ensures that the joint velocities are not too high, where the parameter $\mu > 0$ determines the admissible joint velocities. We refer to $k_P$ and $\mu$ as *control* and *optimality* parameters, respectively.

Our model posits that humans compute their joint velocities by performing a gradient descent on $J$, i.e., $\dot{u} = -\eta\nabla_u J(e_x, u) + \xi_u$, where $\eta > 0$ is the *inverse learning rate* and $\xi_u$ is the *exploratory noise*, modeled as white noise with intensity $\sigma_u$ that captures the inaccuracies in the computation of the gradient and the exploration by humans in the joint velocity space. Upon simplification, the $u$ dynamics are

$$\dot{u} = -\eta((\Phi^\top \hat{W}^\top \hat{W}\Phi + \mu I)u - k_P\Phi^\top \hat{W}^\top e_x) + \xi_u. \tag{8}$$

Putting everything together, the model of human motor learning dynamics

$$\dot{\delta q} = -a\delta q + u + \xi_q \tag{9a}$$

$$\dot{\hat{W}} = \gamma\epsilon\delta q^\top \Phi^\top \tag{9b}$$

$$\dot{e}_x = -W\Phi u \tag{9c}$$

$$\dot{u} = -\eta((\Phi^\top \hat{W}^\top \hat{W}\Phi + \mu I)u - k_P\Phi^\top \hat{W}^\top e_x) + \xi_u, \tag{9d}$$

where $e_x$ dynamics are used instead of $x$ dynamics.

## Results

We first establish the stability and convergence properties of the proposed HML model. We summarize our results in this section and refer interested readers to S1 File:Section 2 for detailed proof.

For initial forward mapping estimate $\hat{C}$ sufficiently close to the actual mapping $C$, and inverse learning dynamics sufficiently faster than forward learning dynamics, the model converges to a neighborhood of the equilibria at which the model learns the true weights $W$ associated with the mapping matrix, the cursor reaches the target position, and the hand posture is

**Table 1. Model parameters.** Fitted parameter values for the 6 subjects.

| Parameters | | Subject 1 | Subject 2 | Subject 3 | Subject 4 | Subject 5 | Subject 6 |
|---|---|---|---|---|---|---|---|
| $\gamma$ | Forward learning rate | 0.0664 | 0.0030 | 0.0456 | 0.1398 | 0.0013 | 0.1252 |
| $\eta$ | Inverse learning rate | 3.1742 | 3.1448 | 1.5383 | 1.9856 | 2.4916 | 0.7131 |
| $\mu$ | Optimality parameter | 2.4581 | 3.3056 | 3.3072 | 3.5735 | 3.5382 | 3.9744 |
| $k_P$ | Control parameter | 1.3098 | 1.5965 | 3.2714 | 1.8976 | 1.5569 | 2.2515 |
| $\sigma_u$ | Exploration noise intensity | 0.8764 | 1.0165 | 1.0082 | 0.9556 | 1.9749 | 0.9298 |
| $\sigma_q$ | Perceptual noise intensity | 0.1370 | 0.5451 | 0.0508 | 0.0169 | 0.7118 | 0.0064 |
| $a$ | Perceptual recency parameter | 10 | 10 | 10 | 10 | 10 | 10 |

stationary. Moreover, the size of the neighborhood is a function of and decreases with the exploration noise intensity.

## Comparing HML model performance with human experiment data

We simulated the proposed HML model with the parameters obtained by fitting experiment data from six healthy human participants to the HML model. The fitted parameters are shown in Table 1 and the fitting procedure is detailed in Materials and Methods. The trajectory data from the HML model was re-sampled at the same time indices as the finger joint angle data from the data glove for direct performance comparisons. We use Forward Model Error (FME) [10], defined by

$$\text{FME} = \frac{\|C - \hat{C}\|_2}{\|C\|_2},$$

as the metric to quantify the convergence of the human estimate of the forward mapping matrix, $\hat{C} = \hat{W}\Phi$, to the actual forward mapping matrix $C = W\Phi$. The two output performance metrics we used to compare the performance of the HML model are reaching error (RE) and straightness of trajectory (SoT). RE in each trial of the human participant data is calculated as the Euclidean norm of the cursor position from the target at the end of the trial. The end of the trial is defined as the instant when the cursor position did not change by more than 0.0025 units for 15 consecutive samples, or 2 seconds after the start of the movement, whichever is earlier. SoT is defined as an aspect ratio of the maximum perpendicular distance of the trajectory from the straight line joining the start and end points, to the straight line distance between the start and the end points.

Fig 1 compares the RE and SoT, and Fig 2 compares the cursor trajectories obtained from the human data with that from the fitted HML model. Results show that our model can reproduce the motor learning of human hand motions very closely. Noisy trajectories, in the beginning, are due to high exploration noise in the initial trials and the fact that the initially learned mapping leads to much of the energy being expended towards the null space of the mapping matrix; trajectories look straighter as the trials progress and the model learns the mapping. The decrease in FME as a function of trials for the subjects (Fig 3) captures the task learning evolution for the human participants.

## Comparative analysis of HML model's efficacy in explaining the motor learning

Pierella *et al.* [10] also developed a computational model of motor learning for high-dimensional episodic/discrete tasks. The extension of a model from capturing a trial-by-trial motor learning

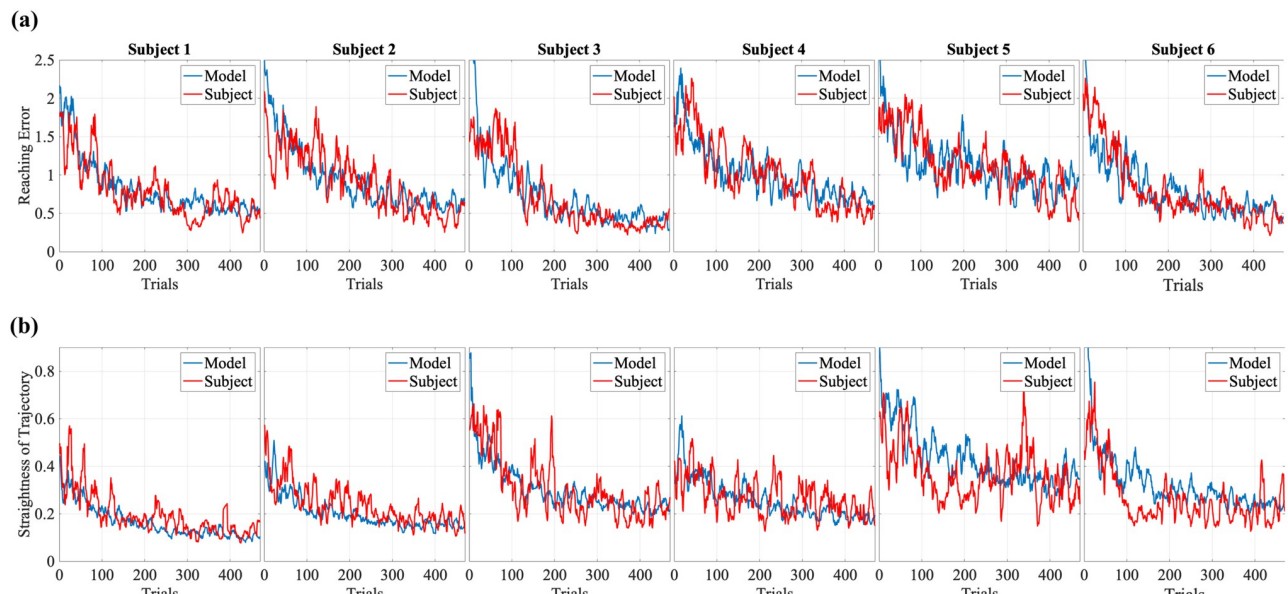

**Fig 1. Performance measures across subjects.** Temporal evolution of (a) reaching error, and (b) straightness of trajectory (both averaged over a moving window of 10 trials) for subjects (red) and the respective fitted HML model (blue) across trials.

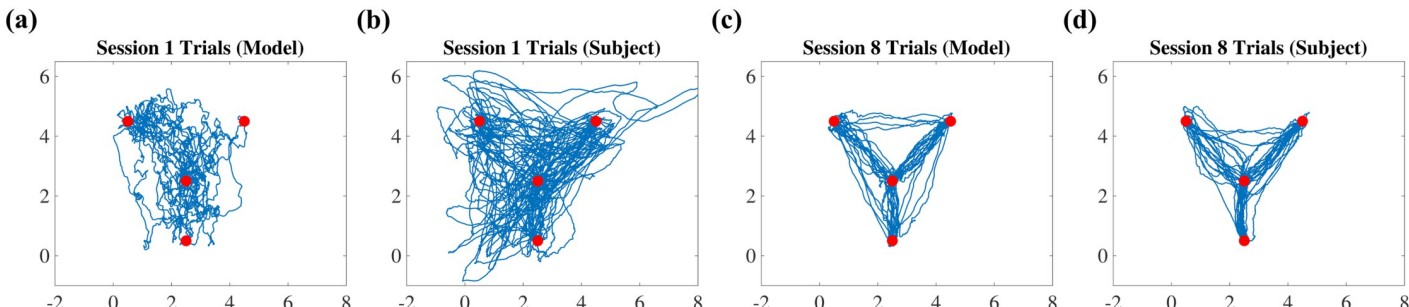

**Fig 2. Cursor trajectories.** Cursor trajectory data from the fitted model (a), (c) and human experiments (b), (d). As learning progresses through the 8 sessions, the trajectories become closer to a straight line between targets, which the proposed HML model also captures.

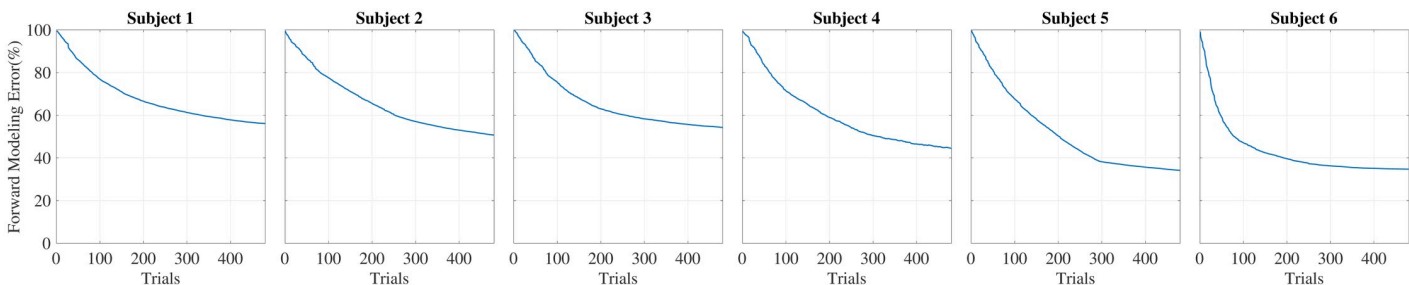

**Fig 3. Forward model error.** Evolution of forward model error (FME) for the fitted model as a function of trials for all 6 subjects.

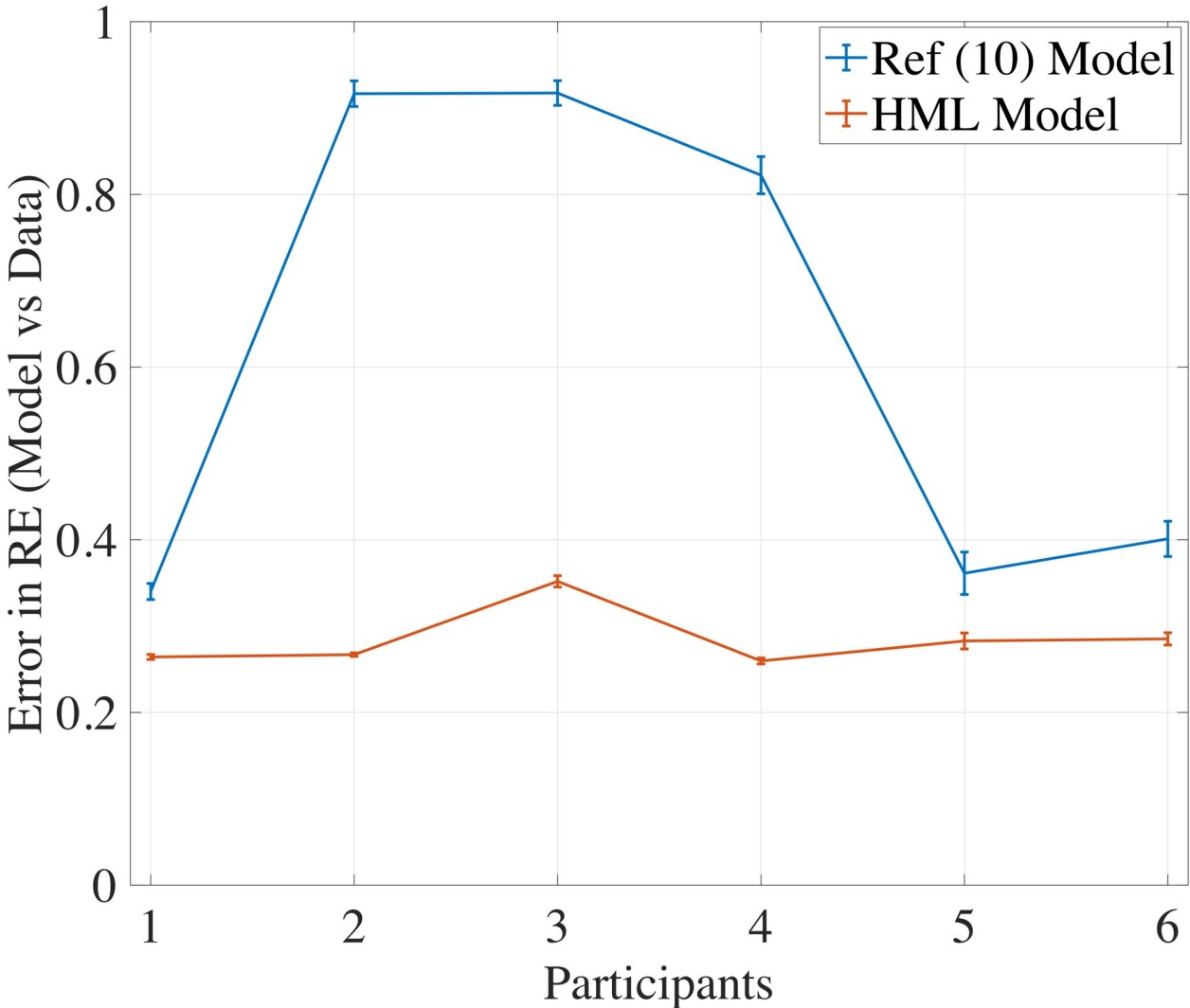

**Fig 4. Comparing HML model with Ref [10] model.** Comparing the errors in RE curve fitting from the model in Ref [10] to the HML model shows that the model in Ref [10] is not as accurate as HML model in capturing the RE for this motor learning task.

task to a continuous task with visual feedback is non-trivial. Nevertheless, we fit the model proposed in Ref [10] and the proposed HML model to the experiment data. For both the models, we report the RE fitting error for all subjects, which captures the deviation of the model's RE from the subject's RE from the experiments in Fig 4. We claim the following improvements over [10] in designing a normative model of human motor learning. First, model fitting errors in Fig 4 show a superior performance of the HML model in capturing the output performance of participants. Second, since the HML model captures continuous learning dynamics, it is possible to extract/estimate a larger set of performance metrics, such as straightness of trajectory [12]. Furthermore, our model introduces a human perception model to account for the continuous visual feedback and incorporates the ideas of motor synergies, capturing prior knowledge about a participant's motor movement behavior. The extracted motor synergies also provide a principled way of initializing the mapping matrix for the model evolution, where the weights on the synergies $W$ can be initialized instead of initializing the whole $C$ matrix randomly.

### Investigating trade-offs in motor learning behavior

We showed that the HML model can capture the human motor learning behavior in novel learning tasks for high-dimensional motor systems very closely. We now conduct a model-based investigation into the influence of parameter variations on various trade-offs in motor learning behaviors. Keeping all the other parameters at their fitted values (Table 1), the parameters under study are varied in a range, and their effects on the performance metrics are observed. For brevity, we only discuss the effects of parameters that had a significant effect on performance metrics.

**Exploration versus exploitation trade-off.**   Motor learning requires balancing the trade-off between the exploration of joint space to learn desired coordinated movement versus the exploitation of the currently learned dexterity to drive the cursor motion. We define the driving (exploitatory) and exploratory efforts as the projection of joint velocities **u** (Eq (9d)) on row space versus null space of (learned) mapping estimate $\hat{C}$ (Eq (9b)), respectively. A typical target-reaching trajectory is assumed to comprise *ballistic* and *learning* phases [24]. In the ballistic phase, the learned model is used for finger joint movement generation, and post this phase, movement corrections based on the visual feedback (learning phase) begin. To better capture the learning of the forward mapping, we compare these two efforts at the end of the *ballistic phase* of each trial. The end of the ballistic phase is calculated at the point within the trial where the norm of finger-joint velocities is maximum.

We found that the inverse learning rate $\eta$ has the most pronounced effect on the exploration-versus-exploitation trade-off. An increase in inverse learning rate in the inverse dynamics (Eq (9d)) initially decreases the energy expended towards the null space of $\hat{C}$, thus increasing the driving effort. Simultaneously, for smaller values of $\eta$, exploration dominates exploitation, and thus the magnitude of joint velocities' projection on null space of $\hat{C}$ is higher. Distribution of these efforts across trials shown in Fig 5 captures this effect, showing a decrease in the mean exploratory effort and an increase in mean driving effort with an increase in $\eta$.

Additionally, for $\eta$ higher than its fitted value in Fig 5, running one-tailed paired t-tests on the distribution of driving effort values across trials between $\eta$-value pairs shows a significant (at significance level $p < 0.001$) difference between the distributions at $\eta$ smaller than its fitted value. The difference between the distributions plateau as we increase $\eta$ past its fitted value. These trends and data fit suggest that human motor learning balances the exploration-exploitation trade-off by tuning $\eta$ such that driving effort is optimally expended towards the task at hand.

**Speed versus accuracy trade-off.**   We define the speed of the response based on the time it takes to capture the target after the start of the movement for each trial. The target is considered captured if the cursor position does not change by more than 0.0025 units for 15 consecutive samples while being inside a radius of size $\rho_x$ around the target point. The accuracy of the response is defined by looking at the straightness of the cursor motion trajectory between the targets, or in other words, the root-mean-square (RMS) error between the cursor trajectory and a straight line joining the start and end targets in a trial.

We found that the control parameter $k_P$ has the most influence on the speed-accuracy trade-off. An increase in the control intensity should lead to faster trajectories and thus a decrease in the average trial time. Additionally, as $k_P$ increases, the driving effort should start dominating the exploration noise, and thus the trajectories should start to look straighter. Fig 6 captures this behavior, where the speed and accuracy go up as $k_P$ increases to a certain value. Comparing the accuracy values across trials around the value of $k_P$ from the data fits (1.3098) to other values of $k_P$ using one-tailed paired t-test shows a significant increase (at the significance level $p < 0.001$) in accuracy values around the fitted $k_P$ value. This may be an outcome of the motor system optimizing this speed and accuracy trade-off for constrained inverse learning effort.

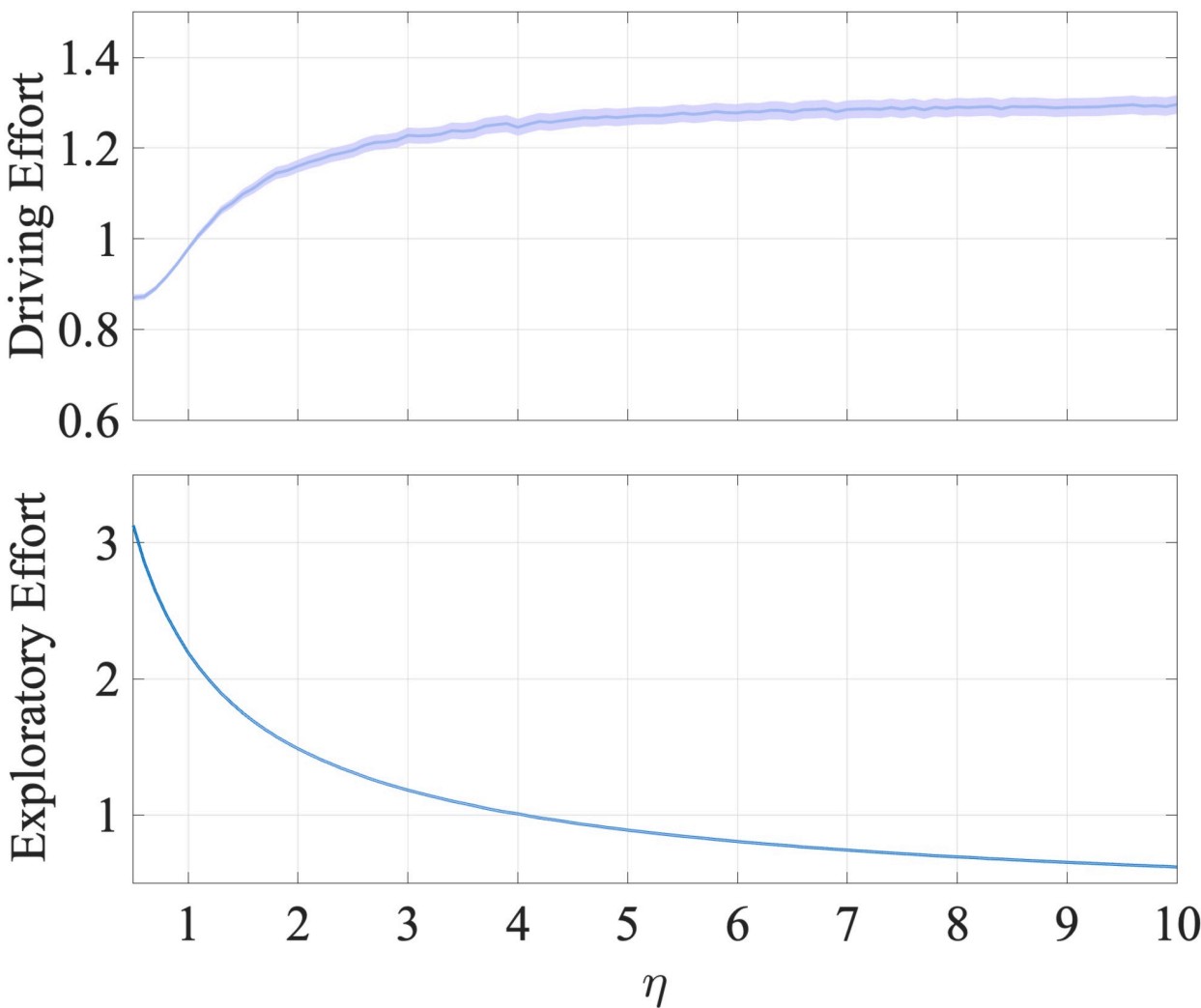

**Fig 5. Effort variation with $\eta$.** Distribution of driving and exploratory effort (averaged across 128 Monte Carlo runs) with means and 95% confidence bounds across trials as $\eta$ is varied around its fitted value 3.1742. While driving effort increases, exploratory effort decreases initially, and both plateau past the fitted $\eta$ value. One-tailed paired t-tests over the effort values across trials reveal this plateauing effect at a significance level of p < 0.001.

**Satisficing.** Satisficing stands for satisfaction and sufficing [6]. In the context of human decision making it refers to the fact that humans tend to settle with 'good enough' options than seeking the optimal ones. In our context, this behavior can be seen in terms of how well the mapping is learned relative to the target size. We model the sufficing level by turning off the learning after a desired threshold is achieved on the Forward Modeling Error (FME), which captures how well the weights on synergies, $W$, have been learned. Satisfaction is modeled using a desired upper bound on the reaching error achieved for a fixed trial time, which is effectively captured by setting the target radius, denoted by $\rho_x$. We thus aim to study how the learning behavior changes by changing these thresholds.

To study satisficing, we quantify the performance with the probabilities with which the trajectories enter different target sizes $\rho_x$ at the end of the prescribed trial time for different learning thresholds (different lower bounds to FME), at different trial times. Intuitively, for a particular learning level (FME threshold), larger target sizes (higher $\rho_x$) lead to better performance

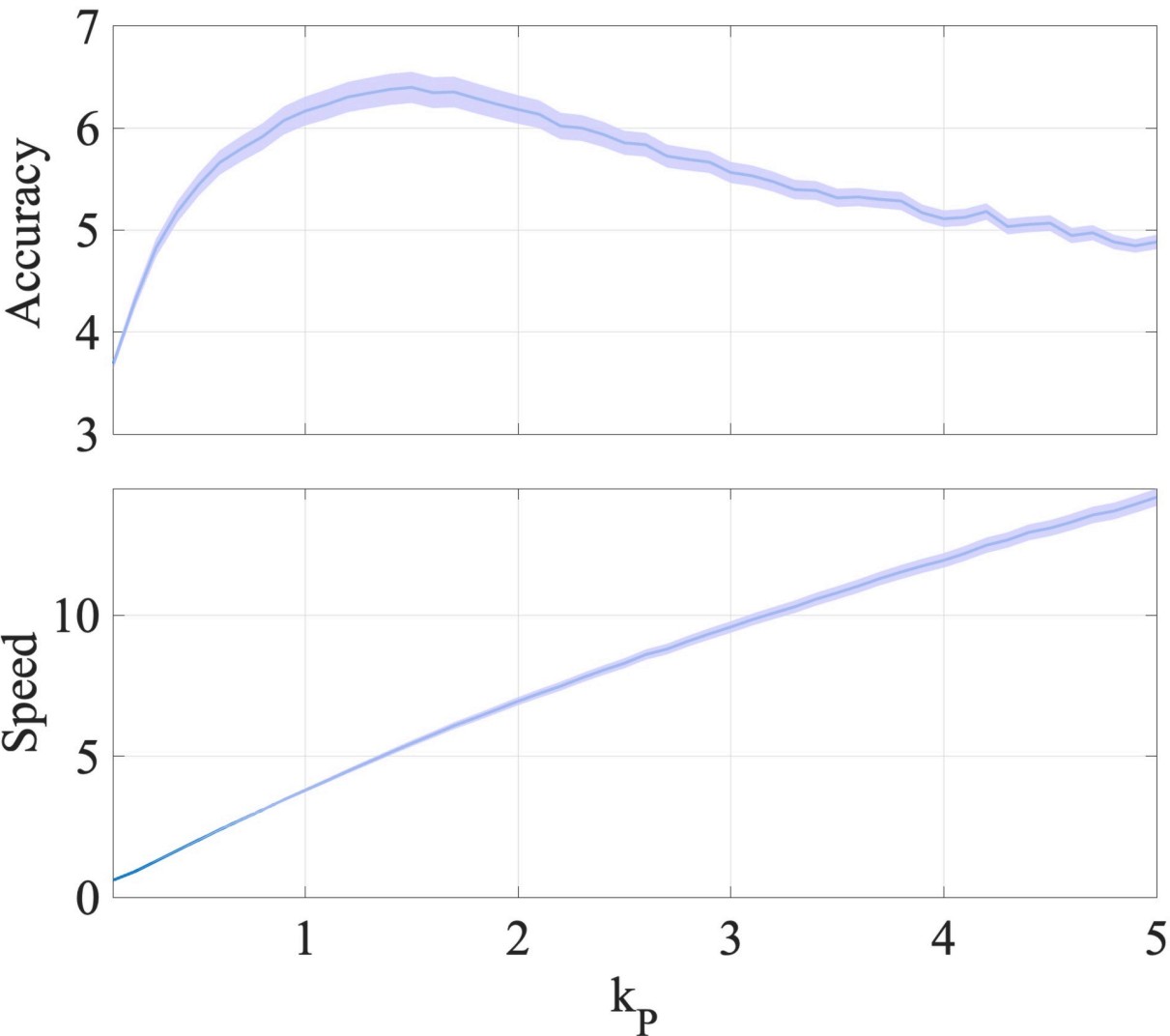

**Fig 6. Speed and accuracy variation with $k_P$.** Across trial distribution (averaged over 128 Monte Carlo runs) of speed and accuracy with means and 95% confidence bounds as $k_P$ is varied around its fitted value 1.3098. Accuracy is highest around the fitted value (p < 0.001) and past that speed increases while accuracy decreases.

(increase success probability). Also, lower FME thresholds are needed at small trial times compared to higher trial times to achieve a particular performance level. This is supported by the curves in Fig 7, which show the probabilities with which the trajectories hit different target sizes, $\rho_x$, for increasing FME threshold across different trial times. Satisficing behavior is observed as curves start to plateau at low FME values, that is, lower values of the learning threshold, which enable better learning, do not necessarily improve the success probability.

Moreover, the learning curves appear to be sigmoid functions; the slope of the sigmoid decreases as the target size is decreased for each trial time. Also, the inflection point starts to move towards higher FME values as both the target sizes and trial times increase. Furthermore, for lower satisfaction levels (higher $\rho_x$ values), the highest success probability does not correspond to the highest sufficing levels (lowest FME thresholds), meaning, enforcing better learning could end up hurting the overall task performance. This effect is more pronounced at smaller trial times.

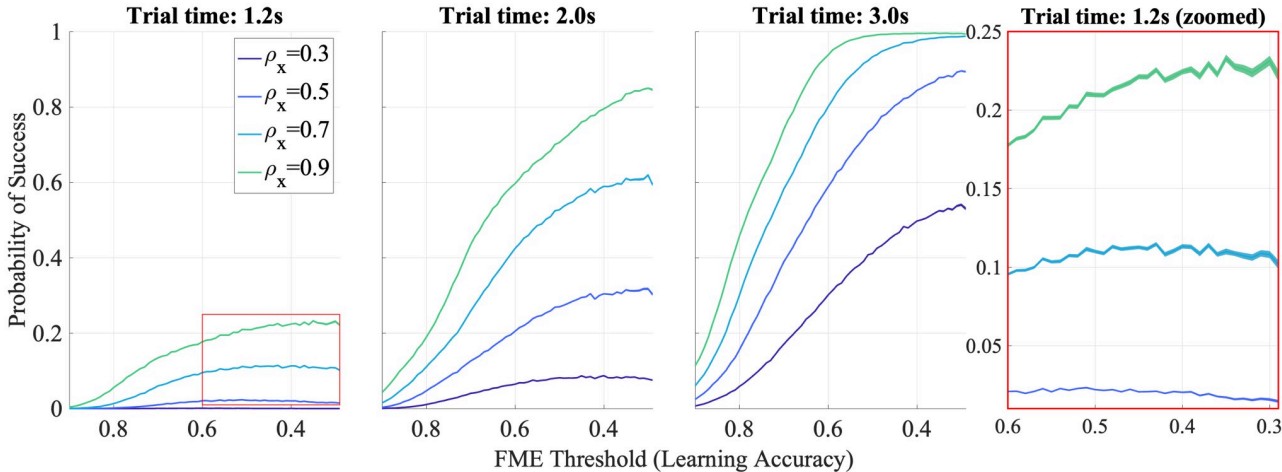

**Fig 7. Satisficing effect.** Probabilities of entering the targets as a function of target size and learning threshold (FME) for different trial times. For smaller trial times, lower learning thresholds (high learning accuracy) are required to achieve high success probabilities for the same target sizes. Satisficing behavior is observed at high learning accuracy (low FME) levels, where learning with higher accuracy does not necessarily increase success probabilities. The zoomed view (right) for probability curves at high learning accuracy (low FME) levels for 1.2s trial time shows the satisficing effect. Curves are average success probabilities with 95% confidence bounds over 1280 Monte Carlo runs of the HML model. Values of $\rho$ are relative to the size of the unit cell of the game grid.

**Flexibility versus performance.** We now explore the effect of the number of motor synergies used in the model on the forward modeling error when the BoMI mapping matrix does not belong to the span of the selected synergies. The higher the number of used synergies, the better an arbitrary mapping matrix can be represented in the space of synergies. However, there is a trade-off as learning with a higher number of synergies requires higher exploratory effort captured through exploration noise intensity $\sigma_u$. This is corroborated by results in Fig 8, which shows the variation of FME with $\sigma_u$ and the number of synergies at the end of session 4. We see minima in FME at synergies less than 19 (the maximum number of synergies) across all exploration noise intensities. Thus, using a higher number of synergies can lead to convergence to the true mapping matrix $C$, however for limited learning time and limited exploration, using a smaller number of synergies gives better performance, as convergence can be much faster albeit not to the true $C$ matrix. This effect is more pronounced around the fit value of $\sigma_q = 0.1370$; for $\sigma_q = 0.1$ using 10 synergies gives us the lowest FME.

## Discussion

The aim of this work is two-fold: to design a computational model of human motor learning for a *de-novo* (novel) learning task that requires participants to capture targets on a computer screen through hand-finger motions, and then leverage this model to study various motor learning trade-offs taking place when learning high-dimensional motor tasks. Toward developing a computational model for human motor learning in a high-dimensional continuous learning task, we tackle the issue of high motor DoFs by leveraging motor synergies extracted from the human participant to create low-dimensional learning representations and include a perception model to account for the continuous visual feedback during the motor task. We then utilize the internal model theory of motor learning to obtain a computational HML model that comprises fast and slow varying forward and inverse learning models. We also establish the exponential convergence properties of the proposed learning model using singular perturbation arguments. Then, we fit the experimental data from 6 human participants to

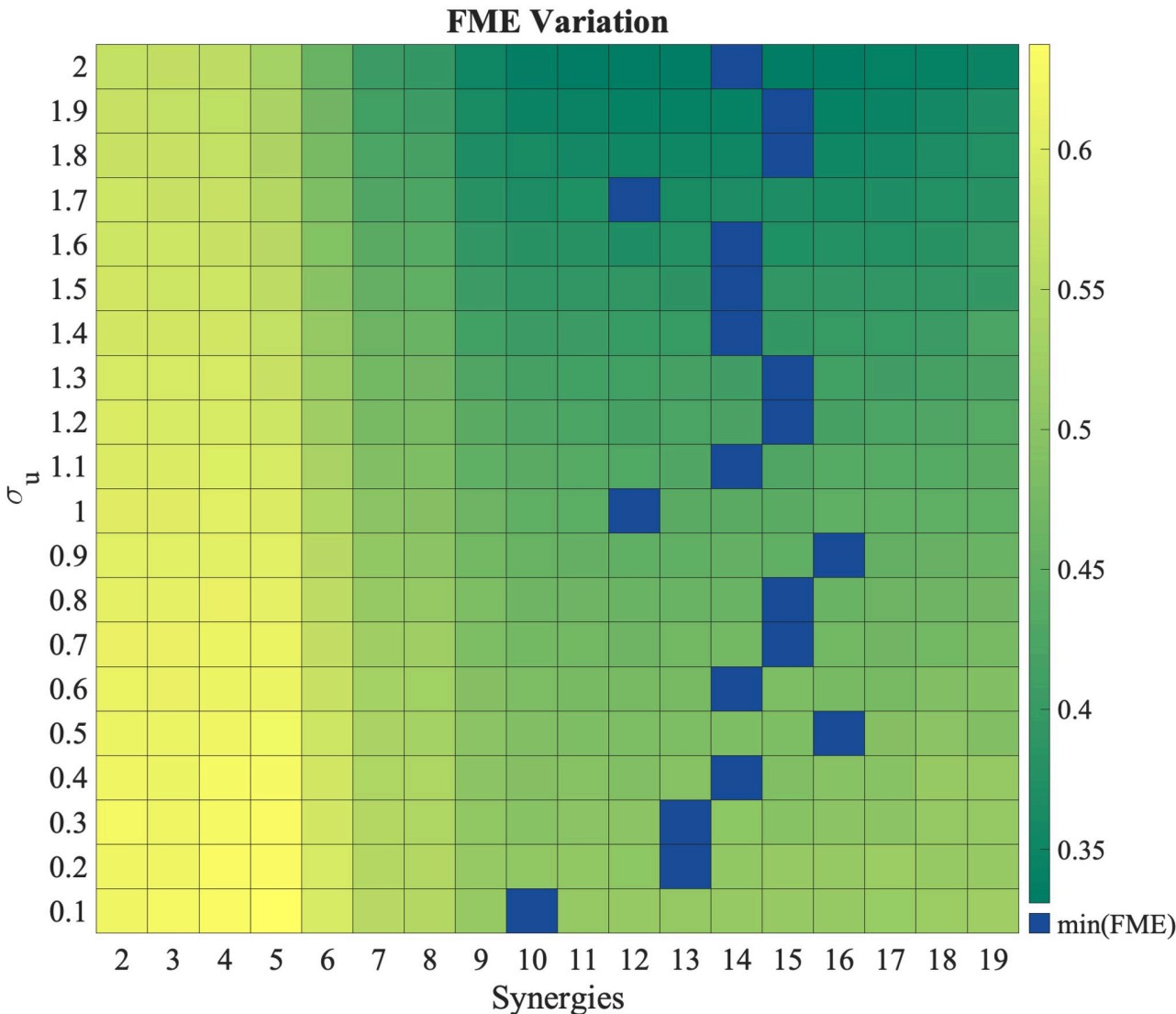

**Fig 8. FME variation with $\sigma_u$ and number of synergies.** FME as a function of increasing $\sigma_u$ and number of synergies used. For limited training time, using more synergies is not always the most optimal strategy. Minimum FME (blue cells) is achieved at synergies lower than 19.

the HML model showing that the proposed model captures the human motor learning behavior well. We then use the proposed model to study the following trade-offs in human motor learning: exploration-exploitation, speed-accuracy, satisficing, and flexibility-performance.

## Motor synergies to capture high-dimensional motor learning

There has been strong evidence of the use of postural synergies by the human nervous system to control high-dimensional motor systems. Literature is replete not only with studies explaining a large number of human hand postures using a small number of postural synergies through kinematic recordings [7, 18] but also with works verifying the encoding of synergistic information for hand postural control at the neural level in human brain motor cortical regions [25–27] and using fMRI brain data to predict the hand postures. It is only natural that the participants would employ these existing motor synergies while learning to play our target

capture game, and thus incorporating synergies in our proposed HML model helps explain the underlying motor learning process more closely. It should be noted however that our model can also capture learning in tasks where the participants have to learn non-synergistic coordination patterns (for example, flexion of thumb accompanied by extension of fingers) simply by not factorizing the mapping matrix into weights and synergies and using $C$ as is.

## HML model and its parameters

Studies in motor learning literature have shown that motor learning in the target capture task cannot be accounted for by mechanisms of motor adaptation, but by de-novo motor skill learning, which consists of developing a controller from scratch, without interfering with pre-existing controllers. Although there are some theories on what the motor learning process for de-novo learning might look like, for example, the new controller could be assembled through reinforcement learning [28, 29], in this work we entertain the possibility of de-novo learning taking place through the formation and simultaneous updating of internal models, including that of perception, and forward and inverse model, which was first introduced in [17].

The coevolution of forward and inverse models of motor learning has had a strong footing in the literature [16, 30–32], [33] [Chapter 9]. However, it has mostly been studied for motor adaptation tasks; care must be taken when generalizing these ideas of dynamically evolving internal models from motor adaptation paradigms to motor skill learning [1]. Nevertheless, few works do explore the coevolution of these internal models in de-novo learning task paradigm [10, 34], of which, [10] came up with a computational model that could explain motor skill learning in an episodic de-novo learning task. We develop a normative model of de-novo motor learning through gradient descent on two separate error functions [15] for continuous feedback tasks. These error functions are constructed using the sensory prediction errors for the forward model [35, 36], and a weighted combination of the task error (a proxy for motor error) and control energy (to account for input energy expended) for the inverse model [17, 37]. If we take out the perception model and the regularization of task error function from our model, we exactly recover the model in [10] for trial-by-trial learning tasks. Of course, when going from an episodic task domain to one with continuous feedback during trials, we need to have some notion of motion perception by human participants. In line with the theories of motion perception in humans, which elucidate that the visual system detects motion by spatio-temporal correlation between the stimuli [38], we design our perception model as smoothing of (cursor and finger-joint) velocities over signal history to obtain the filtered increments in these signals. This perception model includes the perceptual recency parameter $a$ which controls the weights assigned to the previous velocities, and the perceptual noise $\xi_q$ that accounts for inaccuracies in the filtering process. The motivation for including the optimality parameter $\mu$ in the inverse learning model was derived from the likely assumption that a substantial cognitive component is involved in optimizing the motions when learning de-novo tasks. Danziger *et al.* [39] concluded that, as opposed to participants moving the cursor on the screen along straight lines to achieve the task if the cursor is represented as the endpoint of a two-link arm, participants tend to minimize the distance in terms of joint angles of the two-link arm. The second objective that minimizes the total change in two-link arm joint angles during a target capture can be included in the cost function (7) to derive the inverse learning dynamics. The corresponding parameter value $\mu$, that weighs the trade-off between the task objective (capturing the target with robot end-effector) and minimizing the change in two-link arm joint angles, would capture this effect observed in the study. A higher value of the optimality parameter would thus correspond to the participants trying to minimize the total joint angle

changes during a target capture more strongly. However, further investigation is warranted to support this hypothesis, which is beyond the scope of the current work.

## Motor exploration and exploratory noise

Through the motor learning behavior investigation, we identify the parameters that have the most significant effects on the metric trade-offs for the learning task considered in this work. Sternad [40] emphasized the importance of variability in learning novel motor skills by exploring a host of possible solutions in the motor space. This effect is captured by the parameters $\eta$ and $\sigma_u$ in our model. The exploration versus exploitation trade-off analysis (see S1 Fig) shows that an increase in the exploration noise intensity $\sigma_u$ increases the exploration effort, and consequently, the exploration in the nullspace of $\hat{W}$ (equivalently $\hat{C}$) for possible solutions, making FME converge faster. Whereas an increase in $\eta$ has an opposite effect on exploration, while simultaneously increasing the driving effort towards exploitation of the learned policy. We, therefore, hypothesize that the human motor learning system tries to optimize these parameters, deviations from which result in a non-optimal distribution of exploration and driving efforts when learning a motor skill.

## Fast and slow learning timescales

Although not a central focus of our work, we also found evidence for multiple time scales of learning. Two separate time scales of learning, fast and slow processes, in motor adaptation tasks have been widely studied in the literature. Prevailing theories [36, 41, 42] suggest that the cerebellum is involved in the initial fast learning followed by changes in the motor cortex during the slow learning phase [43]. We found a timescale separation between the forward and inverse learning dynamics, which was also corroborated by the parameter fits where $\gamma \ll \eta$. This suggests that in our proposed model, the forward learning dynamics evolves at a slower rate as learning the forward mapping is a process that continues over the whole course of the task. On the other hand, the inverse learning dynamics evolves at a faster rate because the participant has to quickly adapt to generate the optimal finger joint velocities which are computed as per the current (participant) estimate of the forward mapping, thus allowing the participant to complete the trial most optimally. However, a careful evaluation needs to be undertaken to verify this hypothesis.

## Future directions

Having a computational model of HML in high-dimensional motor systems is crucial to advancing our understanding of the underlying learning mechanisms, generating testable hypotheses, guiding the design of effective interventions, and studying the effect of practice schedules, task complexities, and feedback. There have been many studies and experiments in the last decade that aim to design and develop exoskeletons for studying hand joint motions [44], as well as rehabilitation of hand injuries [45]. Recent works [46, 47] have developed assist-as-needed controllers for the rehabilitation of hand fingers. Another work [48] has shown how adaptation in training tasks can modulate the rate of motor learning and affect rehabilitation. The proposed model is capable of explaining motor learning behaviors more broadly, and we believe the experimental paradigm we used is quite extensive for several reasons. Firstly, the experiment involves learning to control a high number of degrees of freedom mapped to a low-dimensional task, which is a critical factor in many motor learning tasks. Secondly, the use of kinematic "synergies" for hand movements has been well documented [7],

enabling the task to capture various features of the model effectively. We plan to test the generality of model on a broader set of learning paradigms going forward.

Future works include leveraging the developed HML model and the insights drawn for the design of such assist-as-needed controllers and adaptive training schedules toward optimizing task performance and/or learning. Furthermore, we can generate hypotheses to experimentally validate various motor learning trade-offs. For example, our model suggests the satisficing behavior during motor learning. This can be empirically evaluated by varying the target sizes $\rho_x$ and examining the relationship between the success probabilities and learning threshold levels. Various metrics can be used as a measure of learning threshold level of a participant, such as straightness of trajectory, reaching error, etc. Another prediction of the model is that on a constrained learning time, using lower number of synergies should lead to a more robust learning performance. A possible experiment could be to evaluate the minimum number of synergies employed by a particular participant to learn the task if we keep perturbing/changing the mapping matrix periodically during the experiment. Specifically, performing PCA on the collected data, we can identify the number of dominant synergies as a function of variation in the mapping.

## Methods

### Experiment procedure

The human participant data for the current paper is from one of the groups from a previous study [12], specifically the group with full visual feedback. The experiment procedure is briefly outlined below, but for more details interested readers are referred to [12].

An experimental session lasts approximately forty-five minutes and consists of three stages.

*Calibration phase:* In this phase, we perform calibration to design the forward mapping matrix specific to each participant. Participants are asked to perform a sequence of free finger movements where they would move their hand fingers in as many different ways as possible, carefully avoiding any extreme ranges of motion, while wearing the data glove. The corresponding posture information is recorded until $4000 - 5000$ samples are collected ($\sim 70 - 90s$), centered around mean posture, and then PCA is performed. The first two principal components (PCs) are used in the mapping matrix $C$ to map the movements of hand finger joints to cursor movement in $x$ and $y$ directions. The two PCs are also scaled by the square root of their respective eigenvalues to ensure comparable ease of motion in two directions. The mean posture is also calibrated to map to the center of the $5 \times 5$ grid.

*Familiarization phase:* In this phase, the participants are asked to move the cursor around freely in a $5 \times 5$ unit grid (for a limited time, so that no motor learning takes place before the training phase) to get accustomed to the game/motions and also to ensure that most of the grid is reachable. The latter is achieved by scaling the game window based on the participant's cursor movement data and it ensures that participants can easily maneuver the cursor across the full screen. Since game window units are calculated from the scaling data during the calibration phase, they are different for different individuals.

*Gameplay (training) phase:* Each participant trains for 8 sessions, each comprised of 60 trials, on 4 target squares (3 outer targets and 1 center target) with centers located at (0.5, 4.5), (2.5, 0.5), (2.5, 2.5), (4.5, 4.5) units on the screen. A trial is comprised of one reaching movement from one target to another. A session always starts after the participants reach the center target, and a target is considered captured if two consecutive glove data samples did not change by more than 2 units ($\sim 1°$) for 10 consecutive samples while being inside the target square. The sequence of targets is randomized in each session, but it always consists of 12 center out movements to each of the 3 outer targets, and 24 movements to the center target (8 from each

each of the 3 outer targets). A score is also displayed at the top of the game window which is calculated based on the accuracy of target capture (proximity to the center of the target square) and time taken to capture the target. Participants are additionally instructed to reach the center of the targets within 2s of the movement start, failing which the target block highlights in red and participants incur a time penalty on the score. This scoring system is shown at the top of the game window which does not serve any functional purpose other than motivating the participants.

### Extracting motor synergies

The matrix $\Phi$ is formed using four motor synergies obtained as the first four principal components from the PCA performed on centered posture data recorded during the calibration phase. While our model can work with any number of synergies, we choose four synergies because previous studies on hand and finger configuration [7, 49] have shown that four synergies spanned more than 80% of the finger joint configurations.

### Fitting HML model to human participant data

We use the human participant experiment data to obtain the HML model parameters in (Eq (9)) (also summarized in Table 1). The task performance is quantified by two metrics—reaching error (RE) and the straightness of the trajectory, to ascertain the performance of the HML model while fitting the data. Reaching error in each trial in the human participant data is calculated as detailed in the Results section. The straightness of the trajectory (SoT) is defined as an aspect ratio of the maximum perpendicular distance of the trajectory from the straight line joining the start and end points, to the straight line distance between the start and the end points. Owing to the stochastic non-linearity of the proposed HML model, we use the multi-objective optimization genetic algorithm NSGA-II [50] to find the optimal parameters over the parameter space using $f_{\text{RE}} = \|\text{RE}_{\text{model}} - \text{RE}_{\text{data}}\|_2$ and $f_{\text{SoT}} = \|\text{SoT}_{\text{model}} - \text{SoT}_{\text{data}}\|_2$ as the two objectives. The subscripts denote if the metric is formed using experiment data or the HML model. The duration of each trial of the HML model was consistent with the human experiment data so as to have a fair basis for objective function calculation. NSGA-II was run for 500 generations using the simulated binary crossover operator and polynomial mutation operator with rates 0.7 and 0.2, respectively, over a population size of 100. Out of the $\min\{f_{\text{RE}}\}$ fits selected from 10 runs of NSGA-II, the parameter fits with minimum $f_{\text{RE}}$ and $f_{\text{SoT}} < \text{avg}\{f_{\text{SoT}}\}$ are chosen for a particular subject. Table 1 shows the parameter values obtained for the human participant experiment data using NSGA-II. Perceptual recency parameter $a$ was heuristically chosen to be a large value $\sim O(10)$, and the target size $\rho_x$ value was chosen the same as the experiments.

### Supporting information

**S1 File. Supporting text.** Document containing supplementary HML model details, its convergence analysis, and additional model-based investigation into human motor learning behavior results.
(PDF)

**S1 Fig. Effort variation with $\sigma_u$.** Distribution of driving and exploratory effort across trials as $\sigma_u$ is varied around its fit value 0.8764. Driving effort is highest around the fitted value of $\sigma_u$ (p < 0.01), while exploratory effort increases monotonically with $\sigma_u$.
(TIF)

## Author Contributions

**Conceptualization:** Ankur Kamboj, Vaibhav Srivastava.

**Data curation:** Rajiv Ranganathan.

**Formal analysis:** Ankur Kamboj.

**Funding acquisition:** Rajiv Ranganathan, Xiaobo Tan, Vaibhav Srivastava.

**Software:** Ankur Kamboj.

**Supervision:** Rajiv Ranganathan, Xiaobo Tan, Vaibhav Srivastava.

**Validation:** Ankur Kamboj.

**Visualization:** Ankur Kamboj.

**Writing – original draft:** Ankur Kamboj.

**Writing – review & editing:** Rajiv Ranganathan, Xiaobo Tan, Vaibhav Srivastava.

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
