## [Decision Letter · Decision Letter 0]

21 Jun 2024

Dear Mr. Kamboj,

Thank you very much for submitting your manuscript "Human motor learning dynamics in high-dimensional tasks" for consideration at PLOS Computational Biology. As with all papers reviewed by the journal, your manuscript was reviewed by members of the editorial board and by several independent reviewers. The reviewers appreciated the attention to an important topic. Based on the reviews, we are likely to accept this manuscript for publication, providing that you modify the manuscript according to the review recommendations.

Sincerely,

Barbara Webb

Academic Editor

PLOS Computational Biology

Andrea E. Martin

Section Editor

PLOS Computational Biology

Reviewer's Responses to Questions

**Comments to the Authors:**

Reviewer #1: This work represents a significant advance in the understanding of motor learning by providing a computational framework that represents continuous learning, a much more ecologically relevant form of learning than the forms of motor learning that have been addressed in countless previous studies. Limitations of the approach include the narrow database used for the empirical application of the model (data from a previously published study) and the use of only one learning paradigm. I found the references to different time scales for learning the inverse model and learning the forward model intriguing. A possible suggestion for improving the manuscript would be to indicate what strong predictions are made for future empirical testing of the framework.

Reviewer #2: The work is defenitely of interest for the scientific community, the model proposed by the authors capable of continuously reproduce the behaviour of subjects while dealing with learning a task in hihgdimensional space is interesting. Despite so I believe that some parts of the paper need to be rearranged and some concepts better explained. Like for example some more details of the model and the experiment used to test it should be introduced earlier in the text to help the reader.

Major.

I like when at the end of the introduction the goal and the hypothesis of the work are presented. Please adjust the text accordingly.

The section "moter learning experiment" should provide a little bit more of details so to allow the user to understand better the results. I know that the methods are at the very end of the manuscript but the reader needs help before the results. For example it is not clear if the task is a center-out task o a random walk. Also, how is the matrix C obtained? it is the same for all participants or each subject has a specific C? Only in the method I read about PCA for example.

Can you please comment about the fact that the subject is controlling the velocity of the cursor and not the position? This is what I understand reading the section about the Model before the Results. But conversely when reading the methods I understand exactly the opposite. There the authors talk about hand postures. What is the subjects controlling? position or velocity? What are the authors modeling? Position or velocity? This part is very important to clarify, because depending on that I might have some additional questions.

line 74 R has dimension hxm...what number is h?

line 105. Why 2 seconds? what is the maximum time available to the subject for reaching a target? please add this detail in the previous part where the task is presented.

It would be nice to see FME curves for all 6 subjects.

line 351-352. Can you please comment more on this hypothesis?

Minor.

In the abstract "Conventional approaches to enhancing..." I believe this should be "to enhance..."

lines 23 and 29, 169 the word "Ref" should not be present, the number between brackets is enough.

Fig 2 caption. I believe the authors used wrong panel labels. From what I see in the Fig. data from human experiment are (b) and (d). Also in the caption the acronym "FME" used in the y label of panel (e) is missing.

Fig 6. What is the unit of measure of ρ?

**Have the authors made all data and (if applicable) computational code underlying the findings in their manuscript fully available?**

Reviewer #1: **No: **I was unable to locate how the authors make their data and code publicly available.

Reviewer #2: Yes

PLOS authors have the option to publish the peer review history of their article (what does this mean?). If published, this will include your full peer review and any attached files.

Reviewer #1: No

Reviewer #2: No

Figure Files:

Data Requirements:

Reproducibility:

References:

---

## [Decision Letter · Decision Letter 1]

4 Sep 2024

Dear Mr. Kamboj,

We are pleased to inform you that your manuscript 'Human motor learning dynamics in high-dimensional tasks' has been provisionally accepted for publication in PLOS Computational Biology.

Best regards,

Barbara Webb

Academic Editor

PLOS Computational Biology

Andrea E. Martin

Section Editor

PLOS Computational Biology

Reviewer's Responses to Questions

**Comments to the Authors:**

Reviewer #1: The authors have addressed all issues raised by the reviewers satisfactorily.

Reviewer #2: I thank the authors for addressing all the concerns I had. For me the work is now ready to be published

**Have the authors made all data and (if applicable) computational code underlying the findings in their manuscript fully available?**

Reviewer #1: Yes

Reviewer #2: None

PLOS authors have the option to publish the peer review history of their article (what does this mean?). If published, this will include your full peer review and any attached files.

Reviewer #1: **Yes: **Joseph Classen

Reviewer #2: No

---

## [Editor Report · Acceptance letter]

23 Sep 2024

PCOMPBIOL-D-24-00667R1 

Human motor learning dynamics in high-dimensional tasks

Dear Dr Kamboj,

I am pleased to inform you that your manuscript has been formally accepted for publication in PLOS Computational Biology. Your manuscript is now with our production department and you will be notified of the publication date in due course.

With kind regards,

Zsofia Freund
